**JCB** Journal of Cell Biology

## TOOLS

# Structure of the F-tractin–F-actin complex

Dmitry Shatskiy[1]* , Athul Sivan[2]* , Roland Wedlich-Söldner[2] , and Alexander Belyy[1]

**F-tractin is a peptide widely used to visualize the actin cytoskeleton in live eukaryotic cells but has been reported to impair cell migration and induce actin bundling at high expression levels. To elucidate these effects, we determined the cryo-EM structure of the F-tractin–F-actin complex, revealing that F-tractin consists of a flexible N-terminal region and an amphipathic C-terminal helix. The N-terminal part is dispensable for F-actin binding but responsible for the bundling effect. Based on these insights, we developed an optimized F-tractin, which eliminates the N-terminal region and minimizes bundling while retaining strong actin labeling. The C-terminal helix interacts with a hydrophobic pocket formed by two neighboring actin subunits, an interaction region shared by many actin-binding polypeptides, including the popular actin-binding probe Lifeact. Thus, rather than contrasting F-tractin and Lifeact, our data indicate that these peptides have analogous modes of interaction with F-actin. Our study dissects the structural elements of F-tractin and provides a foundation for developing future actin probes.**

## Introduction

Actin filaments (F-actin) are an essential component of the cell cytoskeleton involved in numerous intracellular processes, including cell movement, division, and maintenance of cell shape (Dominguez and Holmes, 2011). Due to the critical role of F-actin in cellular morphogenesis, various actin probes have been developed to visualize actin under both physiological and pathological conditions (Melak et al., 2017). These probes include small molecules (Bubb et al., 1994; D'Este et al., 2015), recombinant tags (van Zwam et al., 2024; Martin et al., 2005), actin-binding proteins (Burkel et al., 2007; Schiavon et al., 2020), and peptides (Riedl et al., 2008; Johnson and Schell, 2009). However, the application of these compounds can disrupt normal cytoskeletal homeostasis, making it crucial to identify and understand the molecular basis of any side effects for accurate data interpretation.

Phalloidin, a cyclic peptide derived from the fungus *Amanita phalloides*, was one of the first probes used for F-actin labeling (Wehland et al., 1977; Wulf et al., 1979). This small molecule remains a gold standard for actin visualization due to its high affinity and selectivity for F-actin (Melak et al., 2017). However, its low membrane permeability and the stabilization of actin filaments limit its use to fixed cells (Coluccio and Tilney, 1984). The need to visualize actin dynamics led to the development of genetically encoded fluorescent probes. Chimeras of green fluorescent protein (GFP) with actin have been a simple and popular technique to visualize actin filaments in living cells (Clark et al., 2013; Koestler et al., 2009). However, the bulky GFP tag on actin has been shown to significantly affect actin

assembly (Nagasaki et al., 2017; Aizawa et al., 1997). As alternatives to GFP-actin, GFP-labeled actin-binding domains such as those from utrophin (Burkel et al., 2007) and fimbrin (Sheahan et al., 2004) have been used. In addition, affimers (Lopata et al., 2018) and actin-binding nanobodies (Schiavon et al., 2020) have been developed and successfully used in a variety of cell types and organisms.

To minimize steric clashes of large probes, small fluorophore-labeled peptides have been developed. The most commonly used is Lifeact, a 17-amino acid peptide derived from the N-terminus of the *Saccharomyces cerevisiae* actin-binding protein 140 (Abp140). This probe has been cited in over 1,500 studies, and in the majority of reports, Lifeact did not interfere with (Riedl et al., 2008, 2010; Ignácz et al., 2023) or had minimal effect on the cytoskeleton architecture (Sliogeryte et al., 2016; Belin et al., 2014). However, Lifeact dramatically altered actin-dependent morphogenesis in certain experimental systems, including yeast (Courtemanche et al., 2016), *Drosophila* (Spracklen et al., 2014), *Caenorhabditis elegans* (Ono, 2024), zebrafish (Xu and Du, 2021), and mammalian cells (Flores et al., 2019). A possible molecular mechanism for these cellular artifacts was proposed based on the cryo-EM structure of the Lifeact–F-actin complex (Belyy et al., 2020; Kumari et al., 2020). The structure and subsequent biochemical experiments demonstrated that Lifeact competes with cofilin, myosin, and other actin-binding factors for the same binding site on actin filaments, providing a possible basis for Lifeact-induced artifacts in cell morphogenesis. The availability of high-resolution structural data and the

[1]Membrane Enzymology Group, Groningen Institute of Biomolecular Sciences and Biotechnology (GBB), Faculty of Science and Engineering, University of Groningen, Groningen, The Netherlands; [2]Institute of Cell Dynamics and Imaging, and Cells-in-Motion Interfaculty Center (CiMIC), University of Münster, Münster, Germany.

*D. Shatskiy and A. Sivan contributed equally to this paper. Correspondence to Alexander Belyy: a.belyy@rug.nl; Roland Wedlich-Söldner: wedlich@uni-muenster.de.

pressing need for better probes to visualize and control the actin cytoskeleton inspired the recent development of an optogenetic Lifeact variant (Kroll et al., 2023). In addition, the reports of Lifeact-induced artifacts motivated researchers to consider alternative actin-binding peptides as probes (Sugizaki et al., 2021; Effiong et al., 2024; Ivanova et al., 2024; Nashchekin et al., 2024).

F-tractin, a 43 amino acid-long peptide, is an alternative to Lifeact for the visualization of actin filaments (Johnson and Schell, 2009; Yi et al., 2012). The peptide consists of residues 10–52 of the rat actin-binding inositol 1,4,5-trisphosphate 3-kinase A. F-tractin labels F-actin structures without causing female sterility or significant actin defects during *Drosophila* oogenesis (Spracklen et al., 2014) and has been proven to reliably report on F-actin organization in primary neurons without notable effects on cell function (Patel et al., 2017). However, in

*Xenopus* XTC cells, expression of F-tractin induced radial actin bundles and longer filopodia (Belin et al., 2014). Such bundling activity could also explain the reported increase in cell stiffness and reduction in cell migration associated with F-tractin (Vosatka et al., 2022). Despite the widespread use of F-tractin, the molecular basis for these artifacts is poorly understood due to the lack of structural and biochemical data.

In this study, we determined the structure of the F-tractin-F-actin complex using cryo-EM. The 3.2 Å structure reveals that F-tractin is composed of a flexible N-terminal region and an amphipathic C-terminal helix. By creating an optimized version of F-tractin devoid of the flexible region, we have shown that the N-terminus is dispensable for F-actin binding and that its removal dramatically decreases actin bundling in vitro and increases the exchange rate in FRAP experiments. The C-terminal

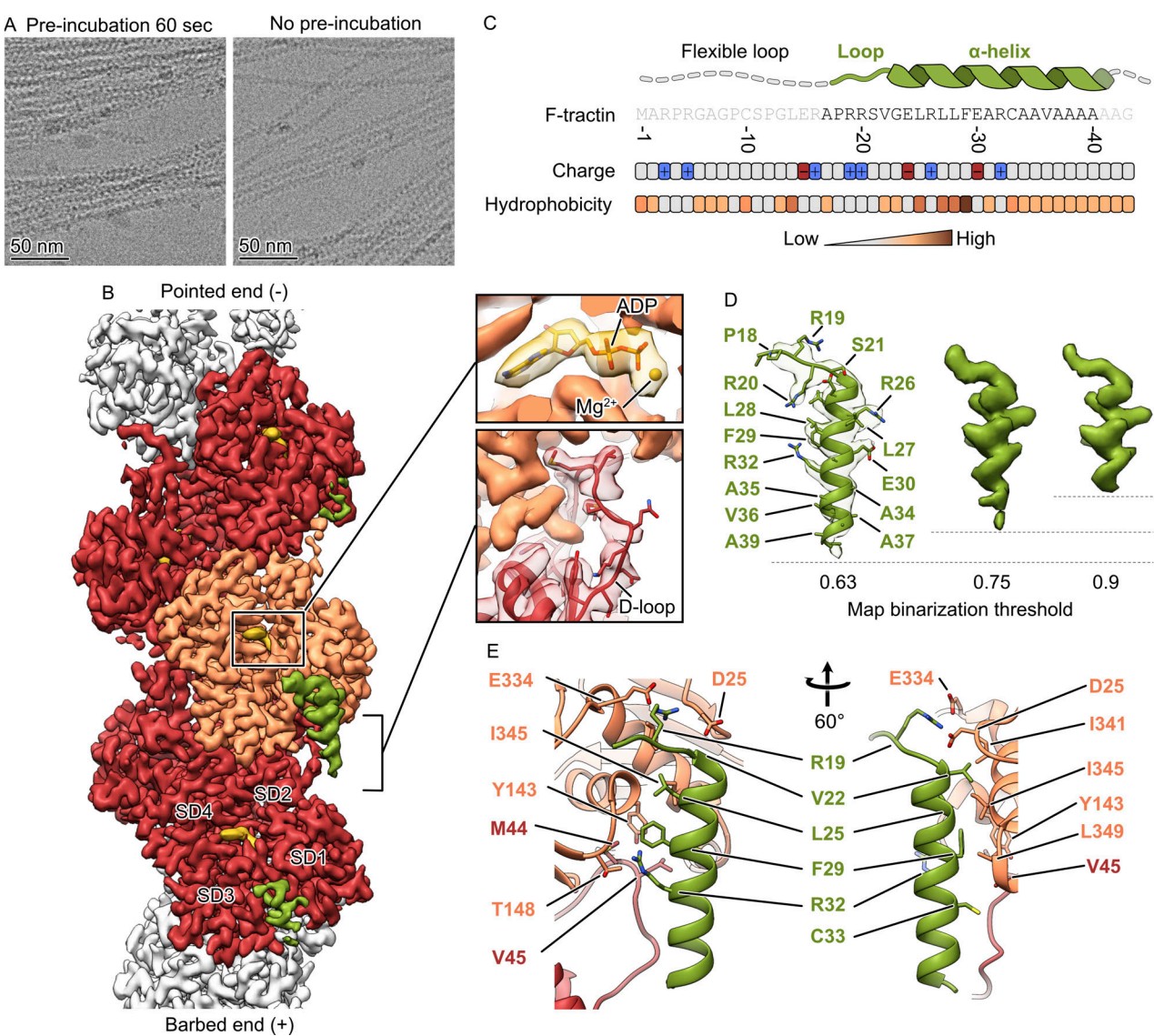

Figure 1. **Cryo-EM structure of the F-tractin–F-actin complex. (A)** Cryo-EM micrographs of actin filaments after mixing with F-tractin with (left) and without preincubation (right). **(B)** The 3.2 Å map of the F-tractin–F-actin complex shows a clear density for actin (orange and red), ADP (gold), and the F-tractin peptide (green). SD – subdomain. **(C)** The amino acid sequence of F-tractin, the distribution of its charged and hydrophobic residues, and the schematic structure of the peptide. **(D)** F-tractin and its corresponding density at different map binarization thresholds, illustrating the flexibility of the extreme C-terminus. **(E)** Atomic model of the F-tractin-F-actin interface.

Table 1. **Cryo-EM data collection, refinement, and validation statistics**

| Dataset | F-tractin-F-actin complex | F-tractin$_{opt}$-F-actin complex |
|---|---|---|
| Microscope | Talos Arctica | |
| Voltage (kV) | 200 | |
| Defocus range (μm) | −0.5 to −2.5 | |
| Camera | Gatan K3 Superresolution mode | |
| Pixel size (Å) | 1.04 | |
| Total electron dose (e/Å²) | 45 | 55 |
| Exposure time (s) | 3.5 | 4 |
| Frames per movie | 50 | 50 |
| Number of movies | 3,391 (8,100) | 3,681 (4,222) |
| 3D Refinement | | |
| Number of particles | 265,256 | 298,240 |
| Final resolution (Å) | 3.2 | 3.4 |
| Helical rise (Å) | 27.4 | 27.4 |
| Helical twist (°) | −166.6 | −166.6 |
| Atomic model statistics | | |
| Non-hydrogen atoms | 14,815 | 14,798 |
| Molprobity score | 1.53 | 1.47 |
| Rama distribution Z-score | −1.88 ± 0.17 | −1.83 ± 0.18 |
| Clashscore | 4.96 | 4.42 |
| Bond RMSD (Å) | 0.003 | 0.003 |
| Angle RMSD (°) | 0.732 | 0.726 |
| Cβ deviations >0.25 Å | 0 | 0 |
| Poor rotamers (%) | 0 | 0 |
| Favored rotamers (%) | 99.62 | 99.62 |
| Ramachandran favored (%) | 96.11 | 96.27 |
| Ramachandran allowed (%) | 3.89 | 3.79 |
| Ramachandran outliers (%) | 0 | 0 |

helix binds to a hydrophobic pocket formed by two actin subunits of the same filament strand. This binding site is similar to that of many other actin-binding proteins, including cofilin, myosin, and the actin probe Lifeact. The overlap in the binding interface suggests competition between F-tractin and actin-binding proteins in cells. Our results help to predict potential experimental artifacts using F-tractin and provide a strong foundation for the development of improved actin-binding probes.

## Results

We produced the 43 amino acid F-tractin peptide using solid-phase synthesis. The peptide was nearly insoluble in water, Tris-buffered saline (TBS), or DMSO, but soluble in methanol at a

concentration of 2 mM. Expecting a micromolar affinity of F-tractin for F-actin similar to Lifeact (Belyy et al., 2020; Kumari et al., 2020), we incubated preformed F-actin with 100 µM of the peptide to obtain a high decoration of actin filaments with the probe. The mixture was then applied onto a grid and plunge-frozen in liquid ethane. Cryo-EM analysis of the sample revealed large heterogeneous bundles instead of single actin filaments (Fig. 1 A). These high-order structures impeded data analysis, prompting us to minimize the interaction time between F-actin and F-tractin. By mixing F-actin with the peptide immediately prior to application to the grid, we observed a sufficient number of individual actin filaments for single-particle cryo-EM analysis. During processing of the dataset, we noticed that the additional density corresponding to F-tractin does not appear at the same binarization threshold as F-actin. Assuming that F-tractin occupancy on F-actin is not 1, we performed alignment-free 3D classification with a focused mask around the peptide to remove actin subunits not containing the peptide. Indeed, after removing 49% of the particles, we obtained a 3.2 Å reconstruction that allowed us to build the atomic model of the F-tractin-F-actin complex (Fig. 1 B, Fig. S1, and Table 1).

F-actin exhibited the expected ADP-state conformation with a closed D-loop (Fig. 1 B). The RMSD value of 0.4 Å between the central actin subunit of the previously published structure of ADP F-actin (Oosterheert et al., 2022) and the central subunit of F-actin in complex with F-tractin indicated that F-tractin did not alter the structure of F-actin. We were able to unambiguously build amino acids 17–40 of F-tractin (Fig. 1, C and D). The 16 N-terminal residues, which include six prolines and glycines, could not be resolved due to their flexibility.

Following the flexible region, the next five residues (APRRS) form a structured loop, with Arg-19 forming a salt bridge with Asp-25 and Glu-334 of actin. The peptide continues as an extended α-helix that spans two adjacent actin subunits along the same filament strand. The interaction interface between the helix and F-actin is primarily hydrophobic: Val-22, Leu-25, and Phe-29 of F-tractin fit into a hydrophobic pocket formed by Tyr-143, Ile-341, Ile-345, and Leu-349 of one actin subunit, and Met-44 and Val-45 of a neighboring subunit (Fig. 1 E). Additionally, Arg-32 forms hydrogen bonds with Thr-148 of actin, further stabilizing the complex. The adjacent Cys-33 faces the actin filament but does not engage in direct interactions. While Cys-33 constitutes a potential site for covalent labeling with sulfhydryl-reactive groups such as maleimides, the close proximity of the attached chemical group to the actin filament could affect the properties of both the filament and the chemical group. Finally, the density quality of F-tractin decreases gradually starting from Ala-34 (Fig. 1 D), suggesting increased flexibility and the absence of specific interactions between the extreme C-terminal residues of F-tractin and F-actin.

The low solubility of F-tractin in aqueous solutions and its strong bundling effect on F-actin in vitro are significant disadvantages that prompted us to improve the peptide based on our structural data. We made three modifications: (1) we removed 16 N-terminal amino acids, including three positively charged arginines, which can form nonspecific interactions with negatively charged actin filaments and be responsible for the

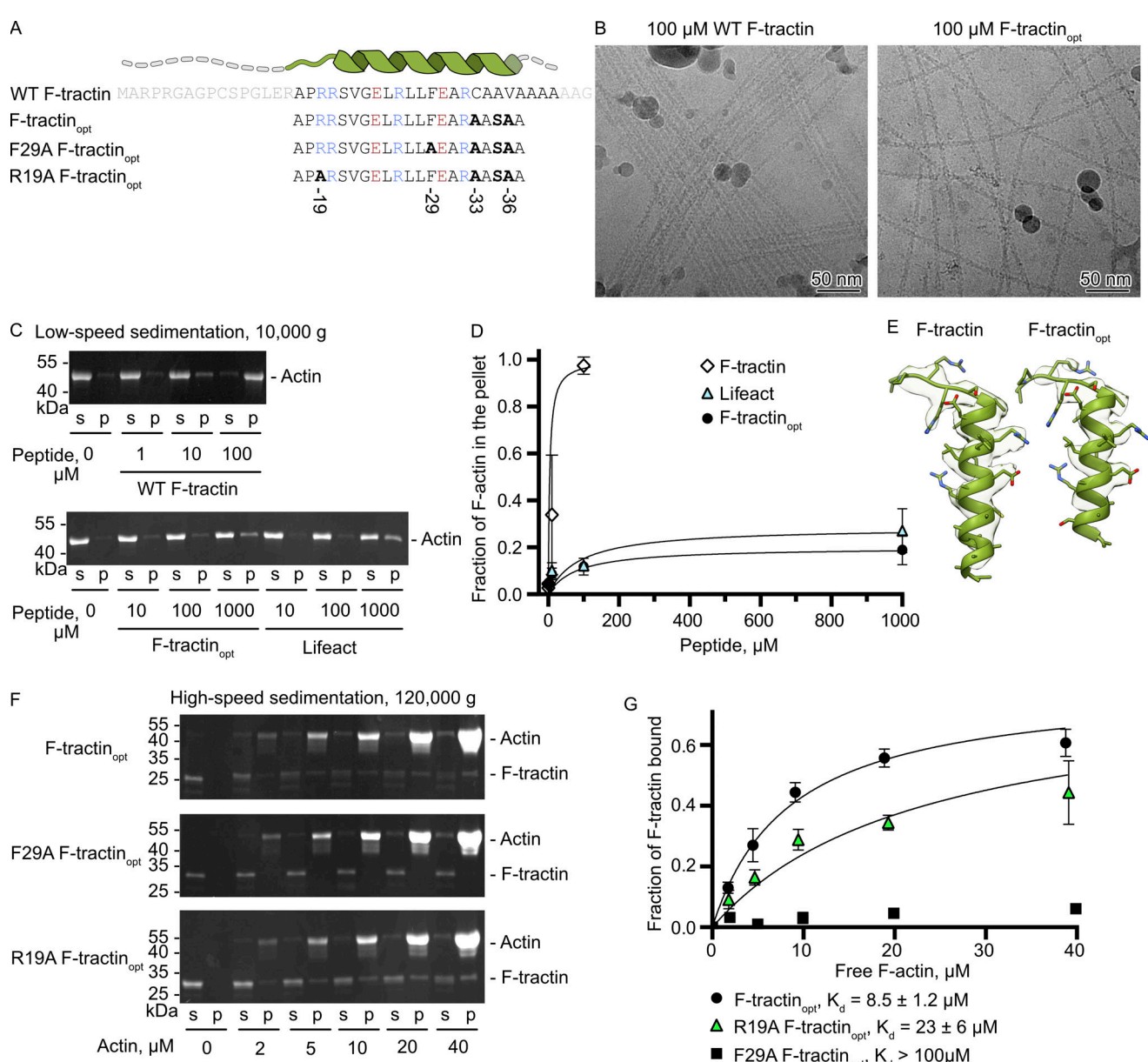

Figure 2. **Point mutations and deletions in F-tractin alter its interaction with F-actin. (A)** Amino acid sequence of F-tractin variants. Amino acids colored in light grey were not observed in the cryo-EM reconstruction. Amino acids colored in blue and red correspond to positively and negatively charged residues, respectively. **(B)** Cryo-EM micrographs of F-actin mixed with 100 µM of WT F-tractin or F-tractin$_{opt}$. **(C)** Actin bundling assay using low-speed centrifugation of F-actin in the presence of WT F-tractin, F-tractin$_{opt}$, or Lifeact. Representative SDS-PAGE from three independent experiments are shown. **(D)** The level of actin bundling was quantified by densitometry and plotted against peptide concentrations. **(E)** Structures of WT F-tractin and F-tractin$_{opt}$ fitted into their cryo-EM densities. **(F)** Cosedimentation of F-actin and 2 µM of F-tractin–mCherry with supernatant (s), and pellet (p) fractions analyzed by SDS-PAGE. The upper band corresponds to actin and the lower band corresponds to F-tractin$_{opt}$–mCherry. Representative stain-free gels are shown. **(G)** The fractions of F-tractin$_{opt}$–mCherry that cosedimented with F-actin were quantified by densitometry and plotted against actin concentrations. Error bars in D and G correspond to standard deviations of three independent experiments. Source data are available for this figure: SourceData F2.

F-actin bundling effect; (2) we mutated cysteine 33 to alanine to prevent the formation of disulfide bonds that can lead to F-tractin oligomerization; (3) we substituted alanine 35 with serine, valine 36 with alanine, and removed the extreme C-terminus containing five alanines and one glycine that do not form interactions with F-actin. The resulting optimized peptide, F-tractin$_{opt}$, was chemically synthesized, water-soluble, and exhibited minimal actin bundling when used at concentrations up to 100 µM, as demonstrated by low-speed centrifugation

assays and cryo-EM (Fig. 2, B–D). We then determined the structure of F-tractin$_{opt}$ in complex with F-actin at 3.4 Å (Fig. S2 and Table 1). The structure of the optimized peptide (Fig. 2 E) and its interaction interface with F-actin was nearly identical to that of the WT F-tractin, confirming that we successfully removed dispensable amino acids without altering the core of the peptide. The lower resolution of the extreme C-terminus of F-tractin$_{opt}$ indicates that the removal of more amino acids at the C-terminus would lead to the loss of the helical structure of the peptide,

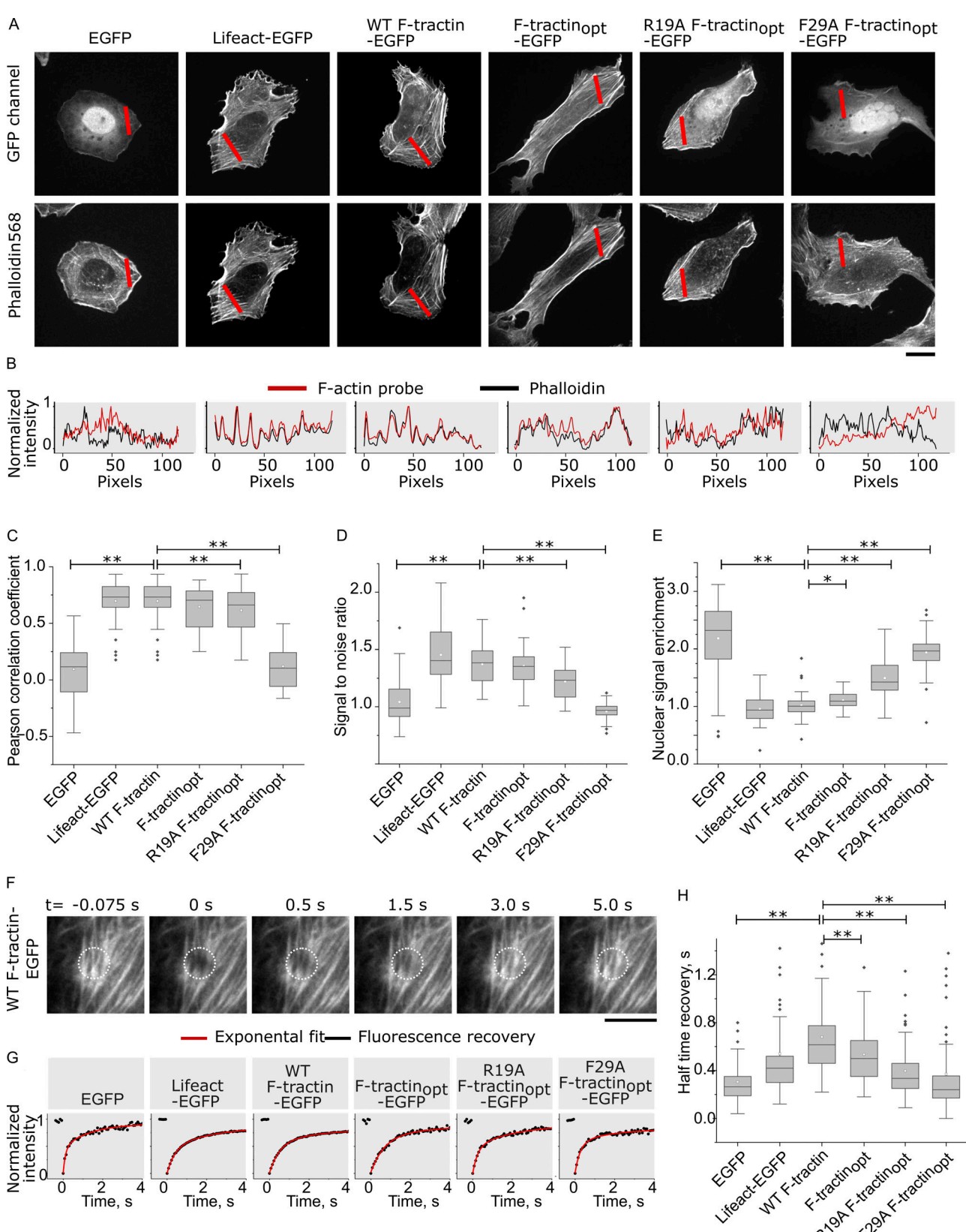

Figure 3. **Actin labeling efficiency and exchange rate of F-tractin variants in U-2 OS cells. (A)** Representative images of F-tractin-EGFP (GFP channel) and phalloidin. Scale bar: 20 μm. **(B)** Line scans of F-tractin variants (red lines) compared with respective phalloidin staining (black lines). **(C)** Boxplots of Pearson correlation coefficients between F-tractin variants and phalloidin staining. Two-sided $t$ test P ≤ 0.01: ** ($N$ = 3 experiments, $n$ > 18 cells). The whiskers extend to 1.5× the interquartile range beyond the first and third quartiles of the data. **(D)** Boxplots of signal-to-noise ratio (fibers/cytosol) for F-tractin variants. $t$ test P ≤ 0.01: ** ($N$ = 3 experiments, $n$ > 27 cells). **(E)** Boxplots of nuclear enrichment for F-tractin variants. Two-sided $t$ test P ≤ 0.01: **, 0.01 < P ≤ 0.05: * ($N$ = 3

experiments, *n* > 27 cells). **(F)** Time frames of a FRAP experiment in U2-OS cells expressing F-tractin-EGFP. Scale bar: 5 µm. **(G)** Recovery profiles for F-actin variants (black) with exponential fits (red). **(H)** Boxplots of FRAP recovery half times for indicated F-tractin variants, free EGFP, and Lifeact. Two-sided *t* test P ≤ 0.01: **. (*N* = 3 experiments, *n* > 25 cells).

disorganization of interactions, and, consequently, a decrease in its affinity for F-actin. These proof-of-principle experiments demonstrate that our structural data provide a robust foundation for engineering actin visualization probes with improved properties.

In live cell imaging experiments, F-tractin is typically used as a fusion of fluorescent proteins. To evaluate the equivalent F-tractin$_{opt}$ probe, we expressed and purified an F-tractin$_{opt}$–mCherry fusion and determined its binding to F-actin using high-speed cosedimentation assays (Fig. 2, F and G). We observed a dose-dependent increase of F-tractin$_{opt}$ in the pellet fraction and estimated the F-actin–F-tractin$_{opt}$ K$_d$ to be 8.5 ± 1.2 µM. Unfortunately, expression of the control mCherry-F-tractin full-length fusion in *E. coli* resulted in an extremely low

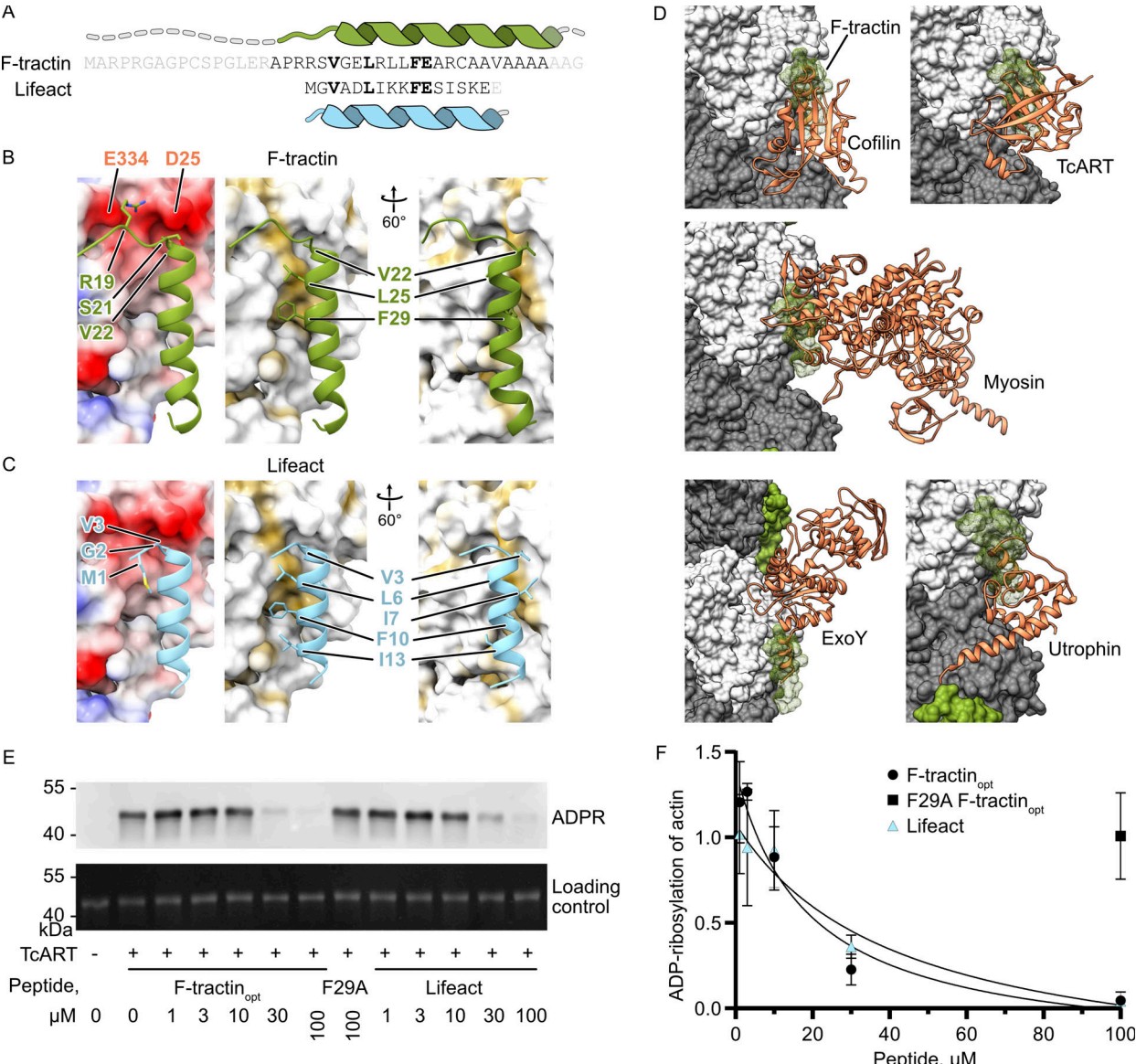

Figure 4. **F-tractin and Lifeact share an interaction interface. (A)** Comparison of amino acid sequence and secondary structure elements of F-tractin and Lifeact. **(B and C)** Comparison of electrostatic (left) and hydrophobic (middle/right) interactions of F-tractin and Lifeact (PDB 7AD9 [Belyy et al., 2020]). **(D)** Structures of cofilin (5YU8 [Tanaka et al., 2018]), myosin (5JLH [Oosterheert et al., 2022]), TcART (7Z7H [Belyy et al., 2022]), ExoY (7P1G [Belyy et al., 2021]), and utrophin (6M5G [Kumari et al., 2020]) bound to F-actin (grey and white). Note that all proteins overlap the F-actin interaction site of F-tractin (green). **(E)** Levels of actin ADP-ribosylation by TcART in the presence of F-tractin$_{opt}$ and Lifeact analyzed by western blot of a His-tagged ADP-ribose binding protein. **(F)** Level of ADP-ribosylation of actin was quantified by densitometry and plotted against peptide concentrations. Error bars in F correspond to the standard deviations of three independent experiments. Source data are available for this figure: SourceData F4.

yield, preventing direct comparison of the affinities between WT F-tractin–mCherry and F-tractin$_{opt}$–mCherry. Nonetheless, the similar K$_d$ of 6.8 µM observed for the longer 66-amino acid-long F-tractin fused to the carrier protein NusA (Johnson and Schell, 2009) suggests that our optimization did not greatly alter the affinity for F-actin.

In agreement with the biochemical data, F-tractin–EGFP and F-tractin$_{opt}$–EGFP showed comparable actin labeling efficiency in U-2 OS osteosarcoma cells—either measured as colocalization with phalloidin (Fig. 3, A–C) or as signal-to-noise ratio between actin fibers and the cytosol (Fig. 3 D). Fluorescence recovery after photobleaching (FRAP) experiments indicated that F-tractin$_{opt}$ exchanged faster than WT F-tractin (Fig. 3, F–H), suggesting a reduced affinity to actin. This was also confirmed by a slightly higher accumulation of F-tractin$_{opt}$ in the nucleus (Fig. 3 E), a characteristic behavior for cytosolic GFP fusion proteins (Seibel et al., 2007).

Having established the use of F-tractin$_{opt}$ as the actin probe, we aimed to determine whether point mutations could further modify the F-tractin$_{opt}$ interaction with F-actin. To this end, we generated two F-tractin$_{opt}$ variants with the following mutations: Phe29Ala, to disrupt the central hydrophobic interaction with F-actin, and Arg19Ala, to eliminate the salt bridge between the central loop and negatively charged residues on the F-actin surface. Cosedimentation assays revealed that these mutants had significantly lower affinity for F-actin, with Phe29Ala F-tractin$_{opt}$ being barely detectable in the pellet at the highest concentration of F-actin used (Fig. 2, F and G). Consistent with the low affinity measured in vitro, R19A F-tractin$_{opt}$ exhibited a significant drop in colocalization with phalloidin (Fig. 3, A–C), reduced signal-to-noise ratio (Fig. 3 D), and higher nuclear enrichment (Fig. 3 E), highlighting its diminished actin-labeling efficiency. In contrast, F29A F-tractin$_{opt}$ completely failed to bind to actin in cells and essentially behaved similarly to free GFP (Fig. 3, A–E). FRAP experiments further confirmed the expected increase in exchange rates (Fig. 3, G and H). These results highlighted the importance of the interactions formed by Arg-19 and Phe-29 of F-tractin$_{opt}$ and demonstrated the potential of structure-based modifications to refine F-tractin interactions with F-actin.

Finally, we decided to directly compare F-tractin with another actin-binding probe Lifeact. Our structural analysis showed that these peptides are analogous and bind to the same region on the actin filament (Fig. 4, A–C). Thus, they should similarly compete with other actin-binding proteins. To test this, we produced the F-actin-binding bacterial ADP-ribosyltransferase effector TcART from Photorhabdus luminescens, which transiently interacts with F-actin at the F-tractin/Lifeact binding site (Belyy et al., 2022). We used TcART to modify F-actin in the presence of different concentrations of Lifeact and F-tractin$_{opt}$ (Fig. 4, E and F). While Phe29Ala F-tractin$_{opt}$ did not interfere with TcART modification of F-actin, both Lifeact and F-tractin$_{opt}$ inhibited TcART activity in a similar dose-dependent manner, confirming that Lifeact and F-tractin$_{opt}$ are nearly identical actin-binding probes.

## Discussion

Since its first description, F-tractin has been considered an alternative to Lifeact for imaging actin dynamics in living cells. However, a comparison of both markers has been largely limited to characterizations in cells (Belin et al., 2014; Spracklen et al., 2014; Vosatka et al., 2022). With structures of both probes in complex with F-actin available, we can now directly compare their actin interactions in atomic detail (Fig. 4, A–C). Both probes are at their core amphipathic helices that bind to overlapping interfaces of consecutive actin subunits, indicating a preference for "aged" ADP-F-actin with a closed D-loop over ATP-F-actin, as previously demonstrated for Lifeact (Kumari et al., 2020). Lifeact forms interactions with F-actin via five hydrophobic residues (Val-3, Leu-6, Ile-7, Phe-10, and Ile-13), while F-tractin only utilizes three of these residues (Val-22, Leu-25, and Phe-29) and instead includes an additional N-terminal loop that forms salt bridges with Asp-25 and Glu-334 of actin. Importantly, mutations of the central phenylalanine (Phe-10 in Lifeact and Phe-29 in F-tractin) completely abolish the interactions of either probe with F-actin (Belyy et al., 2020) (Fig. 2 G). The two probes also exhibit comparable in vitro affinities for F-actin (F-tractin$_{opt}$: 8.5 ± 1.2 µM; Lifeact: 14.9 ± 1.6 µM [Belyy et al., 2020] or 5.8 µM [Courtemanche et al., 2016]) and show similar colocalization with phalloidin and FRAP recovery rates in cells (Fig. 3). Thus, they bind to actin filaments in an analogous fashion and likely compete with other actin-binding proteins such as cofilin, myosin, and bacterial effectors to a similar extent.

Our competition experiments with TcART confirm that Lifeact and F-tractin$_{opt}$ indeed compete with this actin-binding protein in a comparable dose-dependent manner, further indicating their functional equivalence. These results, together with the structural and biochemical data, suggest that the previously reported differences between Lifeact and F-tractin originate from the flexible distal areas of F-tractin or from differential expression levels, rather than from fundamentally different modes of interaction with F-actin. Overall, this work provides crucial information for cell biologists seeking to choose between these actin-binding probes and offers a framework for further engineering actin-visualization tools with improved properties.

## Materials and methods
### Protein production and purification
Rabbit skeletal muscle α-actin was purified from muscle acetone powder (Pel-Freez Biologicals) as described in Merino et al. (2018) and stored in 50 µl aliquots at –70°C.

TcART, the ADP-ribosyltransferase from P. luminescens Tc toxin, was purified from E. coli BL21-CodonPlus(DE3)-RIPL harboring the plasmid pB656 as described previously (Belyy et al., 2022).

Three F-tractin variants fused to mCherry and the his-tagged ADP-ribose-binding protein Af1521 with mutations K35E and Y145R (Nowak et al., 2020) were purified from E. coli BL21-CodonPlus(DE3)-RIPL cells harboring the plasmids pB1043, 1044, 1045, and 791, respectively (Table S1). A single colony was inoculated into 200 ml of LB media and grown at 37°C. At OD$_{600}$ 0.6–1.0, protein expression was induced by the addition of IPTG

to a final concentration of 0.02 mM. After 16 h of expression at 22°C, the cells were harvested by centrifugation, resuspended in TBS (20 mM Tris pH 8 and 150 mM NaCl), and lysed by sonication. The soluble fraction was applied on TBS-equilibrated Protino Ni-IDA resin (Macherey-Nagel), washed, eluted with TBS supplemented with 250 mM imidazole, dialyzed against TBS, and stored at –20°C.

WT F-tractin (MARPRGAGPCSPGLERAPRRSVGELRLLFEA-RCAAVAAAAAAG), F-tractin$_{opt}$ (APRRSVGELRLLFEARAASAA), and F29A F-tractin$_{opt}$ (APRRSVGELRLL**A**EARAASAA) peptides were synthesized by Genosphere, France, with >95% purity.

### Cosedimentation assays
Freshly thawed mammalian α-actin was spun down at 120,000 $g$ for 20 min at 4°C using a TLA-120.1 rotor, and the supernatant containing G-actin was collected. The protein was then polymerized by incubation in F buffer (120 mM KCl, 20 mM Tris pH 8, 2 mM MgCl$_2$, 1 mM DTT, and 1 mM ATP) for 2 h on ice. Cosedimentation assays were performed in 20 µl volumes by first incubating F-actin with F-tractin$_{opt}$-mCherry fusions for 5 min at room temperature and then centrifuging at 120,000 $g$ using the TLA-120.1 rotor for 20 min at 4°C. After centrifugation, aliquots of the supernatant and resuspended pellet fractions were separated on SDS-polyacrylamide gels with 2,2,2-trichloroethanol (Ladner-Keay et al., 2018), visualized under ultraviolet light, and analyzed by densitometry using ImageJ (RRID: SCR_003070). The Kd values were calculated using GraphPad Prism software (RRID:SCR_002798).

### ADP-ribosylation by TcART
A total of 8-µl mixtures of 2 µg (4.8 µM final concentration) actin and actin-binding peptides at specified concentrations were preincubated for 5 min at room temperature in the buffer containing 1 mM NAD, 20 mM Tris pH 8, 150 mM NaCl, and 1 mM MgCl$_2$. The ADP ribosylation reaction was initiated by the addition of 0.2 ng (314 pM) of TcART to the mixture. After 10 min of incubation at 37°C, the reaction was stopped by adding Laemmli sample buffer and heating the sample at 95°C for 5 min. The components of the mixture were separated by SDS-PAGE, blotted onto a polyvinylidene difluoride (PVDF) membrane using a Trans-Blot Turbo Transfer System (Bio-Rad), and visualized using a combination of the His-tagged ADP-ribose binding protein at the concentration of 0.01 mg/ml and HisProbe-HRP conjugate. The level of actin ADP-ribosylation was quantified by densitometry using ImageJ.

### Actin bundling assays
8 µl of actin filaments at 10 µM were mixed with peptides at the specified concentration, incubated for 5 min at room temperature, and spun down at 10,000 $g$ for 10 min at 4°C. Aliquots of the supernatant and resuspended pellet fractions were separated on SDS-polyacrylamide gels with 2,2,2-trichloroethanol (Ladner-Keay et al., 2018) and visualized with ultraviolet light.

### Cryo-EM, data analysis, and model building
Actin filaments were prepared as for cosedimentation assays. Shortly before plunging, F-actin was diluted to 6 µM with F-buffer. 18 µl of 6 µM F-actin was mixed with 1 µl of 2 mM WT F-tractin peptide in methanol or 2 mM F-tractin$_{opt}$ in water and supplemented with 1 µl of 0.4% (wt/vol) of Tween-20 to improve ice quality. 3 µl of sample was applied onto a freshly glow-discharged copper R2/1 300 mesh grid (Quantifoil), blotted for 8 s on both sides with blotting force –15, and plunge-frozen in liquid ethane using the Vitrobot Mark IV system (Thermo Fisher Scientific) at 13°C and 100% humidity.

The datasets were collected using a Talos Arctica transmission electron microscope (Thermo Fisher Scientific) equipped with an XFEG at 200 kV using the automated data-collection software EPU version 2.7 (Thermo Fisher Scientific). Two images per hole with a defocus range of –0.5 to –2.5 µm were collected with a K3 detector (Gatan) operated in super-resolution mode. Image stacks with 50 frames were collected with a total exposure time of 3.5 s (WT F-tractin dataset) or 4 s (F-tractin$_{opt}$ dataset) and a total dose of 45 e⁻/Å² (WT F-tractin dataset) or 55 e⁻/Å² (F-tractin$_{opt}$ dataset). 8,100 movies were collected and 4,048 of them were used for data processing for the WT F-tractin dataset. 4,222 movies were collected and 3,681 of them were used for data processing of the F-tractin$_{opt}$ dataset. After motion correction was performed in Relion (RRID:SCR_016274) version 3.1, the micrographs were imported into Cryosparc for patch CTF estimation. 657 (WT F-tractin dataset) or 569 (F-tractin$_{opt}$ dataset) micrographs were discarded due to excessive drift and suboptimal ice thickness. Filaments from the remaining micrographs were picked using Filament Tracer in Cryosparc with a filament diameter of 75 Å and a separation distance between segments of 52.5 Å. 612,235 (WT F-tractin dataset) or 586,554 (F-tractin$_{opt}$ dataset) particles with 300 px box size were extracted and 2D classified into 200 classes. 523,970 (WT F-tractin dataset) or 576,045 (F-tractin$_{opt}$ dataset) particles were selected for Helix Refine with a helical twist estimate of 166.4° and a helical rise estimate of 27.6 Å. After this 3D refinement, we performed a round of global and local CTF refinements and another helical refinement. To improve the density corresponding to the F-tractin peptide, alignment-free 3D classification with a focused mask around the peptide, filter resolution of 10 Å, and force-hard classification settings were used to separate particles into two classes. The larger class containing the density for the peptide with 265,256 (WT F-tractin dataset) or 298,240 (F-tractin$_{opt}$ dataset) particles was locally refined with the "use pose/shift Gaussian prior during alignment," rotational search extent 5°, and shift search extent 2 Å settings to allow only small rotations and shifts of the reconstruction. The B-factor sharpened output of the local refinement was used for model building.

The structure of F-tractin, predicted in Alphafold 2 (Jumper et al., 2021), and the structure of ADP-F-actin (Merino et al., 2018) (PDB 5ONV) were fitted into the density in Chimera (Pettersen et al., 2004) and refined in Isolde (Croll, 2018). Before deposition, the structures were refined in real space in Phenix (RRID:SCR_014224) (Liebschner et al., 2019) with secondary structure and reference model restraints. The statistics presented in Table 1 were calculated by MolProbity (RRID:SCR_014226) (Williams et al., 2018).

## Cell culture and transfection

U2OS human osteosarcoma cells (# HTB-96; ATCC) were cultured in McCoy's medium (with 10% fetal calf serum [FCS]). The U2OS cell line used in this study has not undergone formal authentication. However, the cell line was obtained directly from ATCC and routinely tested for mycoplasma contamination, with no signs of contamination observed during the study. Cells were seeded at a density of 30,000 cells/cm² in an 8-well plate (Sarstedt) and allowed to grow overnight at 37°C (10% $CO_2$) prior to transfection (Yun et al., 2009). The plasmids of all desired actin markers fused with GFP were transfected using the X-fect transfection reagent (Takara Bio) as per the manufacturer's recommendation. 500 ng of each plasmid was mixed with X-fect reaction buffer to make up a final volume of 50 µl. 0.5 µl of Xfect Polymer was added and vortexed shortly before incubating the mixture at room temperature for 10 min. The mixture was spin-down and pipetted into the 8 wells of the plate. The cells were then incubated at 37°C overnight before imaging.

## Fixing and staining

Cells were fixed and stained with Phalloidin-568 (Promokine). The cell culture medium (McCoy's) was discarded and the cells were washed once with Phosphate-Buffered Saline (PBS). The cells were fixed with 4% paraformaldehyde (PFA) (50 µl per well) for 20–30 min at room temperature and washed twice with PBS. 0.01% Triton X-100 (TX-100) was used to permeate cells (3–5 min at RT), followed by a two-time PBS washing. The cells were incubated at room temperature with 100 nM phalloidin for 20 min and then washed twice with PBS. Cells were then stained with Hoechst (1:5,000 dilution from the original stock [H3570; Thermo Fisher Scientific]) for 15 min, followed by two times of PBS washing.

## Imaging

We imaged the fixed cells with the iMIC system from FEI/Till Photonics equipped with a spinning disk unit (Andromeda) using a 40X lens (0.9 NA; Olympus) and lasers at wavelengths of 405, 488, and 561 nm (Omicron Sole-6). Multichannel images were captured using an EMCCD camera (Andor iXon Ultra 897) controlled by Live Acquisition software (Till Photonics). For photobleaching experiments, we used another iMIC system from FEI/Till Photonics for capturing videos with an Olympus 100× oil immersion lens (1.4 NA). The TIRF angles were adjusted prior to each microscopy session. Live-cell imaging was conducted with cells cultured on glass-bottom plates and maintained in Hanks' Buffered Salt Solution (HBSS) supplemented with 10 mM HEPES (pH 7.4) at 37°C. We used 488 nm (Cobolt Calypso, 75 mW) to excite the EGFP-fused F-tractin constructs in cells. For each cell that was imaged, there were three to four FRAP regions of Interest (ROI) (100% laser for 100 ms) and a normalization ROI (0% laser 100 ms). Movies were acquired with a frame interval of 76 ms, and ROIs were bleached sequentially after 100 frames. The timelapse videos were captured Imago-QE Sensicam camera.

## Quantification of recovery rates from FRAP movies

The FRAP videos were imported into Fiji (RRID:SCR_002285) for analysis. Point FRAP ROIs were expanded to circles with a radius of 20 pixels. We customized the standard Fiji FRAP plugin to obtain the altered FRAP ROIs and the normalizing ROI. The plugin measures the average intensity values for each ROI over a specified number of image slices. We then normalize these intensity values, determine the bleaching frame, and calculate the mean intensity before bleaching. The plugin then fits the normalized FRAP recovery curve using an exponential recovery model and calculates half-time recovery. We used a single-term exponential function as reported in previous studies (Koskinen and Hotulainen, 2014). The half-time recovery values of different variants were compared pairwise to WT F-tractin using the two-sided Student's $t$ test. We assumed the data distribution to be normal but was not formally tested.

## Quantification of Pearson correlation coefficient

From the multichannel images (16-bit) captured using the spinning disk microscope, we segmented the cells and nuclei automatically using Cellpose. We used the standard Cyto2 and nuclei modules of Cellpose for generating cell and nuclear masks (8-bit). Following that, we subtracted the nucleus from the cell mask and performed the correlation between the EGFP and Phalloidin slices of the image. We used *pearsonr* function from the scipy.stats module (RRID:SCR_008058) to calculate the Pearson correlation coefficient and the associated P-value for testing correlation. We relied on the assumption of normality in the data, although we did not verify it with a formal test. We used a two-sided Student's $t$ test for the statistical comparison of the dataset with WT F-tractin.

## Quantification of the signal-to-noise ratio (SNR)

Fiber segmentation was applied to the background-subtracted phalloidin channel (50 pixels rolling ball radius) using Otsu's thresholding to isolate fiber structures of cells after removal of the nuclear area. Both, fibers and nuclear areas were subtracted from the cell mask to obtain the cytosolic background. The SNR was calculated as the ratio of mean fluorescence intensity within the fiber region to the mean signal intensity of the background region (Fig. S3). This analysis was done using Python (3.10.13) with the libraries including scikit image for rolling ball background subtraction, OpenCV for morphological operations and mask manipulations, tiff file for reading multichannel TIFF images, and NumPy for array operations and statistical calculations (two-sided $t$ test).

## Quantification of nuclear signal enrichment

The nuclear and cell mask obtained from the cellpose segmentation of cell expressing actin markers were used to quantify the accumulation of nuclear signal. The enrichment ratio was computed as the mean fluorescence intensity of the nucleus divided by that of the rest of the cell. We used Python (3.10.13)-based analysis using OpenCV, tifffile, Matplotlib, and NumPy libraries. The data were not formally tested for normality. We used a two-sided $t$ test to compare the nuclear enrichment of all tested actin probes against that of WT F-tractin.

## Online supplemental material

Fig. S1 shows processing of the F-tractin–F-actin complex. Fig. S2 shows the processing of the F-tractin$_{opt}$–F-actin complex. Fig. S3 shows the image analysis workflow. Table S1 shows the list of primers and plasmids used in this study. Table S2 shows data points.

## Data availability

The coordinates for the cryo-EM structures of the F-tractin–F-actin and F-tractin$_{opt}$–F-actin complexes have been deposited in the Electron Microscopy Data Bank under accession numbers EMD-51496 and 52289. The corresponding molecular models have been deposited at the wwPDB with accession codes PDB 9GOB and 9HM9. The raw cryo-EM data generated during the current study are available from the corresponding author (AB) on request. The raw cell biology data is available at https://omero-imaging.uni-muenster.de/openlink/rn_V3L1ZGYJH1MF_3402_Tractin%7C_Characterization_images_masks/F_Tractin_Project_Colocalization_SNR/ and https://omero-imaging.uni-muenster.de/openlink/rn_FEQT9AAVGDNQ_3402_Tractin%7C_Characterization_FRAP_videos/. Code used to analyze the data is available at https://github.com/athulsivan/Tractin_Charcaterization.git. Individual data points from the graphs at Fig. 2, Fig. 3, and Fig. 4 are available in Table S2.

## Acknowledgments

We thank Artem Stetsenko, Roman Koning, and Frank Faas for help with cryo-EM data collection and Michiel Punter for maintaining the computing cluster. Cryo-EM data were collected at the electron microscopy facilities of the University of Groningen and Leiden University Medical Center.

This work has been funded by the University of Groningen (A. Belyy) and the German Research foundation (DFG, SFB1009, SFB1348, and SFB1557 to R. Wedlich-Söldner). Open Access funding provided by the University of Groningen.

Author contributions: D. Shatskiy: Data curation, Formal analysis, Investigation, Validation, Visualization, Writing - original draft, Writing - review & editing, A. Sivan: Data curation, Formal analysis, Investigation, Methodology, Resources, Software, Validation, Visualization, Writing - original draft, Writing - review & editing, R. Wedlich-Söldner: Funding acquisition, Project administration, Resources, Supervision, Writing - original draft, Writing - review & editing, A. Belyy: Conceptualization, Data curation, Formal analysis, Funding acquisition, Investigation, Methodology, Project administration, Supervision, Validation, Visualization, Writing - original draft, Writing - review & editing.

Disclosures: All authors have completed and submitted the ICMJE Form for Disclosure of Potential Conflicts of Interest. R. Wedlich-Söldner reported a patent to US8957029B2 issued. No other disclosures were reported.

Submitted: 27 September 2024

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

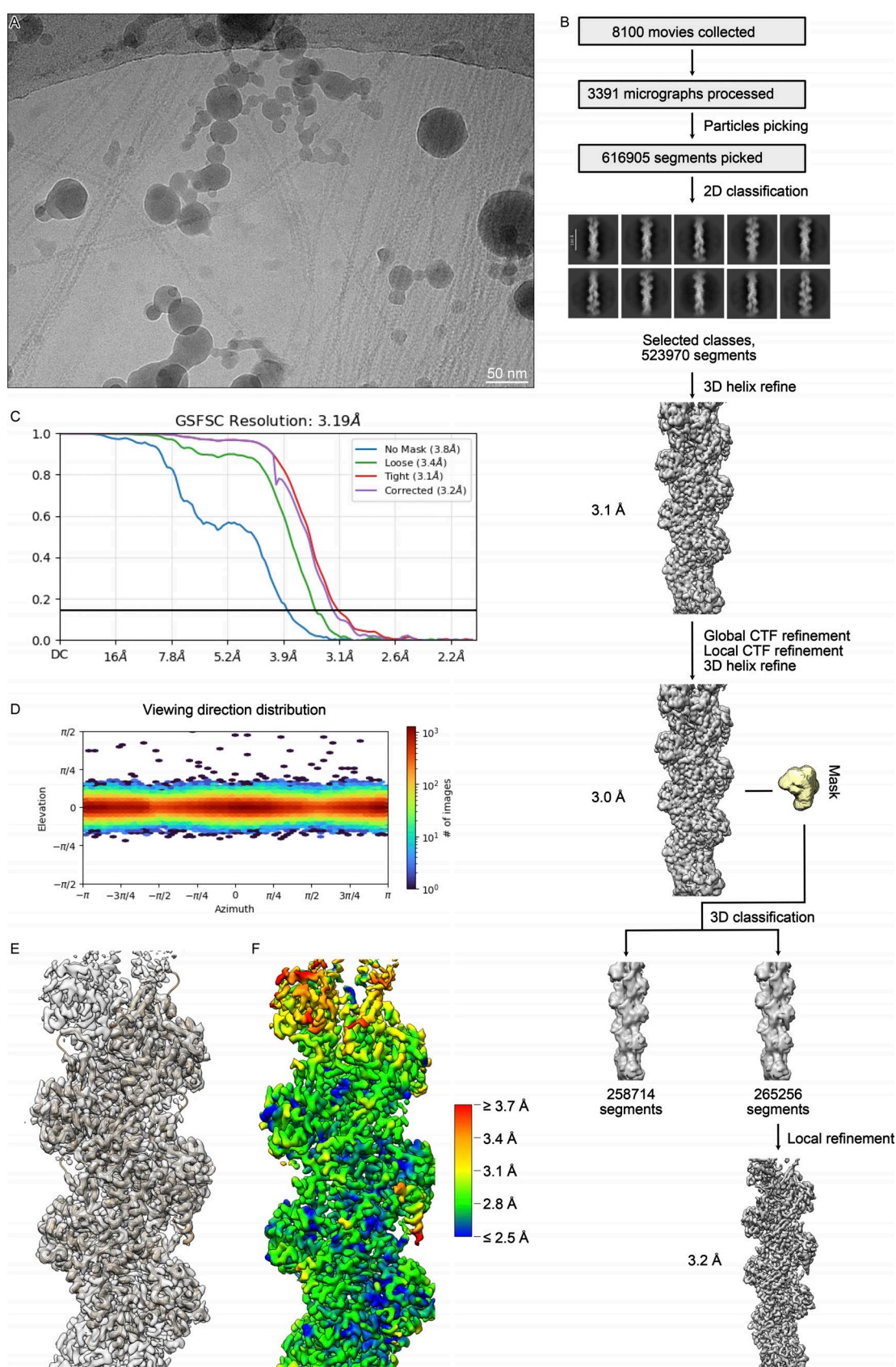

Figure S1. **Processing of the F-tractin-F-actin complex. (A)** An example of the 3,391 analyzed cryo-EM micrographs. **(B)** Processing overview. **(C)** Fourier shell correlation curve of the final reconstruction. **(D)** Angular distribution. **(E)** Fit of the molecular model into the cryo-EM density. **(F)** Local resolution gradient of the reconstruction calculated at the FSC threshold of 0.143.

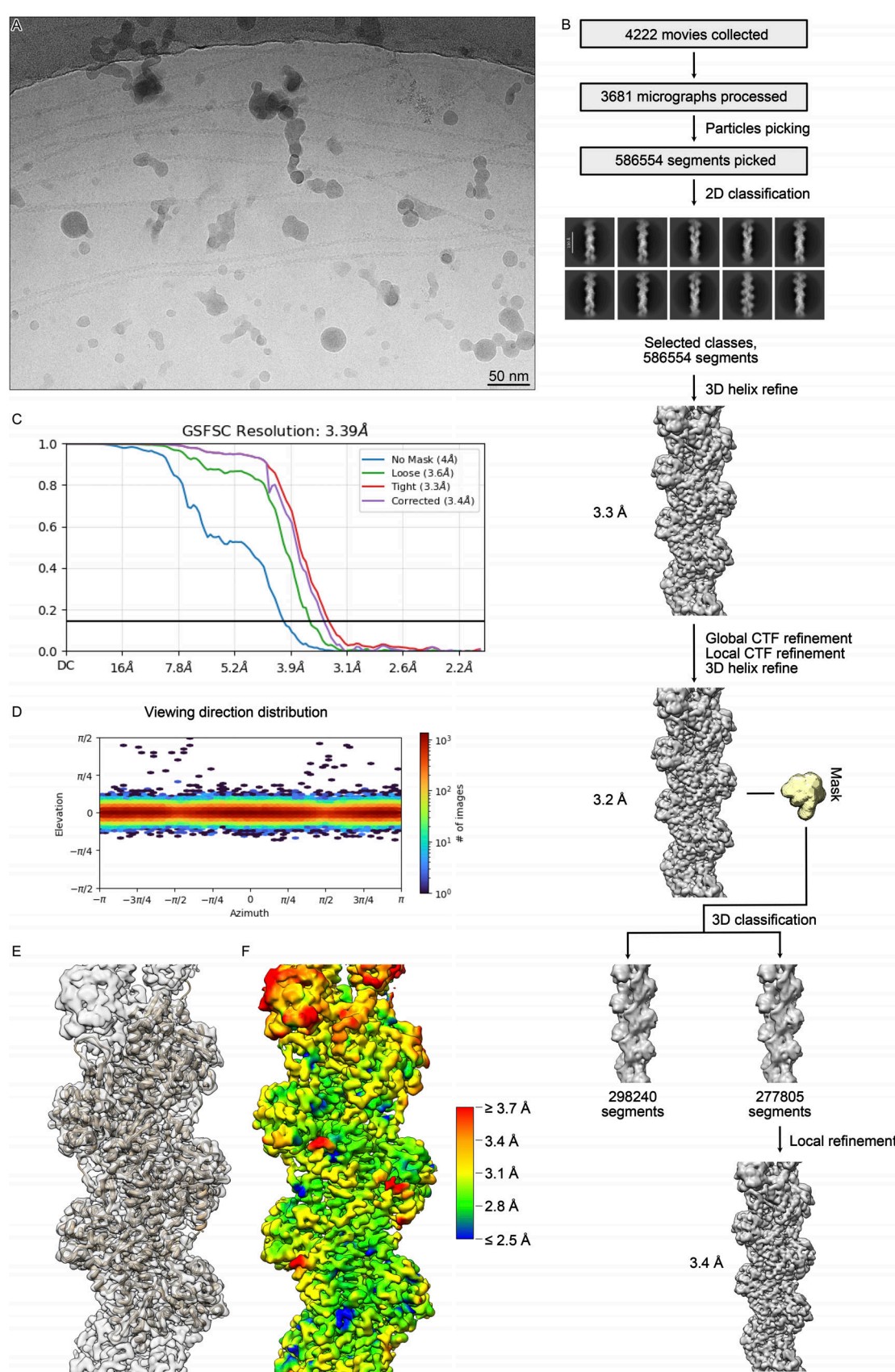

Figure S2. **Processing of the F-tractin$_{opt}$–F-actin complex. (A)** An example of the 3,681 analyzed cryo-EM micrographs. **(B)** Processing overview. **(C)** Fourier shell correlation curve of the final reconstruction. **(D)** Angular distribution. **(E)** Fit of the molecular model into the cryo-EM density. **(F)** Local resolution gradient of the reconstruction calculated at the FSC threshold of 0.143.

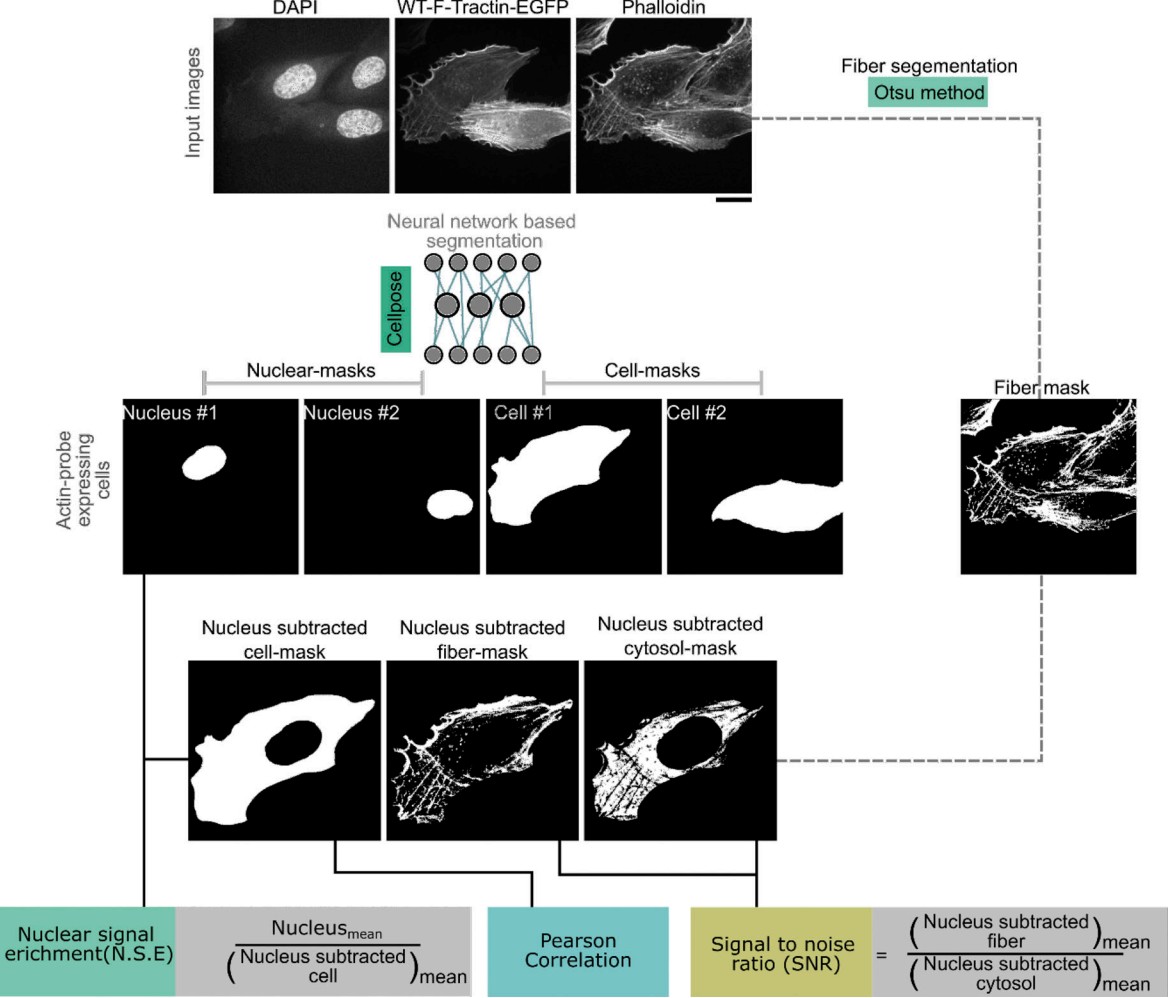

Figure S3.    **Image analysis workflow.** Composite images were segmented using a combination of cellpose-based cell and nuclear segmentation and a custom Python pipeline incorporating background subtraction and Otsu's thresholding for phalloidin channel fiber segmentation. Region-specific intensity measurements were performed for nuclear, fiber, cytosolic, and whole-cell. Pearson correlation coefficients were calculated to assess the relationship between actin markers and phalloidin fluorescence intensities, providing insights into colocalization patterns and enrichment trends.

**Provided online are Table S1 and Table S2. Table S1 shows the list of primers and plasmids used in this study. Table S2 shows data points.**

