## [Peer Review File · The Journal of Cell Biology]

Structure of the F-tractin-F-actin complex

Dmitry Shatskiy, Athul Sivan, Roland Wedlich-Soldner, and Alexander Belyy

Corresponding Author(s): Alexander Belyy, University of Groningen

Review Timeline:

Submission Date:	2024-09-27
Editorial Decision:	2024-11-12
Revision Received:	2024-12-19
Editorial Decision:	2025-01-08
Revision Received:	2025-01-14

Monitoring Editor: Bruce Goode

Scientific Editor: Tim Fessenden

Transaction Report:

DOI: <https://doi.org/10.1083/jcb.202409192>

November 12, 2024

Re: JCB manuscript #202409192

Alexander Belyy
University of Groningen

Dear Dr. Belyy,

Thank you for submitting your manuscript entitled "Structure of the F-tractin-F-actin complex". The manuscript was assessed by expert reviewers, whose comments are appended to this letter. We invite you to submit a revision if you can address the reviewers' key concerns, as outlined here.

You will see that Reviewers 1 and 2 commended the demonstration that F-tractin and Lifeact bind actin filaments in a very similar manner, as well as the potential for the optimized F-tractin shown here. Reviewer 1 made important and constructive points to improve this work, all of which should be addressed in a revision. A suitable revision must respond to all points by Reviewer 2 in some form, however new data requested in points 1-3 from this reviewer are not required. Finally, we felt that Reviewer 3 unfortunately did not appreciate the key conclusions made by this work. We include their remarks for your consideration but we will not consult this reviewer on a revised manuscript.

GENERAL GUIDELINES:

Text limits: Character count for an Tools is < 40,000, not including spaces. Count includes title page, abstract, introduction, results, discussion, and acknowledgments. Count does not include materials and methods, figure legends, references, tables, or supplemental legends.

Figures: Toolss may have up to 10 main text figures. Figures must be prepared according to the policies outlined in our Instructions to Authors, under Data Presentation, <https://jcb.rupress.org/site/misc/ifora.xhtml>. All figures in accepted manuscripts will be screened prior to publication.

Supplemental information: There are strict limits on the allowable amount of supplemental data. Toolss may have up to 5 supplemental figures. Up to 10 supplemental videos or flash animations are allowed. A summary of all supplemental material should appear at the end of the Materials and methods section.

Please note that JCB now requires authors to submit Source Data used to generate figures containing gels and Western blots with all revised manuscripts. This Source Data consists of fully uncropped and unprocessed images for each gel/blot displayed in the main and supplemental figures. Since your paper includes cropped gel and/or blot images, please be sure to provide one Source Data file for each figure that contains gels and/or blots along with your revised manuscript files. File names for Source Data figures should be alphanumeric without any spaces or special characters (i.e., SourceDataF#, where F# refers to the associated main figure number or SourceDataFS# for those associated with Supplementary figures). The lanes of the gels/blots should be labeled as they are in the associated figure, the place where cropping was applied should be marked (with a box), and molecular weight/size standards should be labeled wherever possible.

The typical timeframe for revisions is three to four months. If you anticipate any difficulties in meeting this aforementioned revision time limit, please contact us and we can work with you to find an appropriate time frame for resubmission. Please note that papers are generally considered through only one revision cycle, so any revised manuscript will likely be either accepted or rejected.

Thank you for this interesting contribution to Journal of Cell Biology. You can contact us at the journal office with any questions at cellbio@rockefeller.edu.

Sincerely,

Bruce Goode
Monitoring Editor
Journal of Cell Biology

Tim Fessenden
Scientific Editor
Journal of Cell Biology

Reviewer #1 (Comments to the Authors (Required)):

In their paper "Structure of the F-tractin-F-actin complex", Shatskiy and colleagues present the cryo-EM structure of F-tractin, a commonly used probe for visualizing the actin cytoskeleton with fluorescence microscopy, bound to filamentous actin. Their structure shows that F-tractin binds in a similar manner to another commonly used probe, LifeAct, forming a small amphipathic helix that blocks engagement by other actin binding proteins.

Impressively, the authors use their structural data to generate an optimized version of F-tractin with reduced actin bundling activity, an undesirable feature that can artefactually alter the cytoskeletal dynamics of living cells. They also identify specific residues important for binding, setting the stage for developing improved actin visualization tools.

Overall, this is a short-but-sweet, to the point study which achieves exactly what it set out to do. As optimized actin imaging tools are of broad interest to the cell biology community, I believe this is highly appropriate for the Tool article format in JCB.

I have inspected the cryo-EM map and model provided by the authors, as well as the PDB validation report, and I believe the data and analysis are of very high quality, fully supporting the authors conclusions. I also appreciated that they explicitly displayed the less-resolved regions of the map in the figures (e.g. the base of the F-tractin helix), and commented upon them in the text. For this they are to be commended.

I believe this work is broadly suitable for publication in JCB, pending minor revision to address a few points below.

Minor comments:

- 1) In Fig. 2c, I believe it would be useful to quantify the low-speed bundling assays rather than simply showing a single example gel. These tend to be fairly variable, and analyzing the percent F-actin pelleted across a few trials would provide confidence.
- 2) RE: Fig. 3: one of the important elements of a probe is its signal-to-noise ratio, which in this case would be the degree of apparent F-actin enrichment vs. cytosolic background. If this is similar between F-tractinopt and WT F-tractin, it would be an important "selling point" for the optimized version. One way to measure this would be to quantify the images by making a mask from the phalloidin channel, then calculating the average F-tractin intensity within the mask vs. in the cytoplasm (excluding the nucleus). A scatterplot of these values for the different constructs, or comparing the average ratio of the cytoskeletal vs. cytoplasmic signal, would likely be informative. This analysis could likely be done with the authors' current data.
- 3) In Table S1, the authors list the helical rise as 28.9 Angstroms, but in the Methods the starting value is listed as 27.7 Angstroms (similar to what has been reported in many other F-actin structures). If the rise really jumped 1.2 Angstroms during IHRSR processing, this suggests there is likely a pixel size calibration issue.

Suggestions / typos etc.:

- 4) I believe the potential impact of F-tractinopt is undersold. E.g. it is not really mentioned in the Abstract or Introduction: its potential significance as a useful tool could be emphasized a bit more.
- 5) In the Methods, the authors first say they purified alpha-actin, but then in the co-sedimentation assays subsection they say they used beta-actin. If this is not a typo, the purification of beta-actin needs to be described.
- 6) Line 56: "setups" should probably be "systems"

- 7) Line 81-82: Could mention that the Lifeact binding site is similar here as well.
- 8) Fig. 1a: The contrast on the cryo-EM micrographs was very difficult to see when printed. I'd recommend adjusting the contrast, or low-pass filtering the images for display.
- 9) Line 299: CTF "correction" should probably be "estimation"

Reviewer #2 (Comments to the Authors (Required)):

In this study, Shatskiy and colleagues report a cryo-EM structure of the F-tractin:F-actin complex. The authors resolved 24 out of the 43 amino acids in the F-tractin peptide through their cryo-EM reconstructions. Their findings reveal that the N-terminal part of the F-tractin peptide, while not essential for F-actin binding, is crucial for its bundling effect. Additionally, they identified that the C-terminal helix exhibits interaction profiles similar to several actin-binding proteins. This study highlights that F-tractin and Lifeact comparably bind to the same F-actin sites. The structural insights provided here are instrumental in advancing the design of more effective F-actin probes.

This paper is well-written, conveying a clear and compelling message supported by their robust cryo-EM data. The methodology is appropriately presented. I believe this manuscript will engage the general readership of the JCB journal.

I have the following comments/suggestions:

1. To enhance the study, it would be beneficial if the authors included a discussion on the occupancy of F-tractin in cryo-EM reconstructions. Specifically, an explanation of the peptide's nature and how its solubility impacts its occupancy would provide valuable insights. Furthermore, clarifying how variations in occupancy align with natural binding behaviors could offer a deeper understanding of the interaction dynamics.
2. The authors also created an optimized F-tractin variant, which exhibits no bundling, which is a valuable contribution. Further, solving the complex structure of this optimized F-tractin in complex with F-actin could provide significant insights into bundling/binding. While this is not a hard request, such analysis would help identify essential residues within the C-terminus, distinguishing them from those that are dispensable, thereby enhancing our understanding of its structural and functional roles in binding vs bundling.
3. The finding that F-tractin shares a common interaction interface with actin-binding proteins is intriguing. Strengthening the manuscript further, the authors could perform cellular imaging using the newly designed probes, particularly the optimized F-tractin, to demonstrate co-localization with the interaction partners (any) identified in Figure 4. Given that F-tractin is known to influence actin dynamics and may stabilize actin, it would be valuable to assess whether the novel probes developed in this study offer any functional advantages over native F-tractin.
4. Authors should discuss a possible mechanism for F-tractin mediated bundling. Discuss how it contributes to bundling vs binding in the manuscript.
5. The authors have provided details on fixed cell preparation for imaging; however, it would strengthen the manuscript if they also included a description of the cell preparation protocols used for live cell imaging for conducting FRAP experiments.
6. Expand the model building and refinement section in the methods part of the manuscript.

Reviewer #3 (Comments to the Authors (Required)):

The manuscript by Shatskiy et al entitled "Structure of the F-tractin-F-actin complex" describes the cryoEM structure solution of F-tractin peptide bound to filamentous actin. The authors have validated the structure using structure-function analysis and present a shorter F-tractin peptide called F-tractin-opt, which performs similar to the original peptide. The structure and biochemistry is of decent quality. However, it is hard to comprehend what new findings the authors are presenting, as the title aptly points this manuscript is a structure study and does not offer any insights to cell biologists to mitigate the issues with F-tractin as a F-actin probe or solutions. Therefore, I do not recommend publication in JCB, unless the authors add some new insights using their structure. Few major points are summarized below;

1. Based on structure the authors show that N-terminus is dispensable for F-actin binding and reduces actin bundling. This region is then attributed to the artifacts seen in in vivo and cell studies that uses F-tractin. The authors should show these in vivo methods using F-tractin-opt, this will be a great application of their structure.
2. Similar to the above point, have the authors considered combining lifeact and F-tractin sequence information to produce an alternate, better probe.
3. Since F-tractin binds to similar region as lifeact and it has been shown that D-loop state affects lifeact binding it is not clear

whether D-loop state affects the F-tractin binding. This should be discussed.

4. The authors have used ADP-ribosylation assay as a method to show F-tractin interference; it is not clear what the authors trying to convey here. It should also be noted that, myosin/cofilin and other actin-binding proteins have a larger footprint and may have better binding properties than the TcART.

5. In general the discussion of the manuscript falls very short in many aspects, this needs a complete overhaul.

6. The lifeact Kd indicated in line 207 refers to the author's earlier study. However, other studies have shown different Kd values. This needs to be corrected.

7. Almost all the studies of F29A mutant has been done in the background of F-tractin-opt. The authors should show the mutant effect in original F-tractin peptide.

We thank the reviewers for their positive and constructive feedback, which allowed us to further improve our manuscript. Below we include a detailed response to each point raised.

Reviewer #1

In their paper "Structure of the F-tractin-F-actin complex", Shatskiy and colleagues present the cryo-EM structure of F-tractin, a commonly used probe for visualizing the actin cytoskeleton with fluorescence microscopy, bound to filamentous actin. Their structure shows that F-tractin binds in a similar manner to another commonly used probe, LifeAct, forming a small amphipathic helix that blocks engagement by other actin binding proteins.

Impressively, the authors use their structural data to generate an optimized version of F-tractin with reduced actin bundling activity, an undesirable feature that can artefactually alter the cytoskeletal dynamics of living cells. They also identify specific residues important for binding, setting the stage for developing improved actin visualization tools.

Overall, this is a short-but-sweet, to the point study which achieves exactly what it set out to do. As optimized actin imaging tools are of broad interest to the cell biology community, I believe this is highly appropriate for the Tool article format in JCB.

I have inspected the cryo-EM map and model provided by the authors, as well as the PDB validation report, and I believe the data and analysis are of very high quality, fully supporting the authors conclusions. I also appreciated that they explicitly displayed the less-resolved regions of the map in the figures (e.g. the base of the F-tractin helix), and commented upon them in the text. For this they are to be commended.

I believe this work is broadly suitable for publication in JCB, pending minor revision to address a few points below.

We thank the reviewer for the very positive feedback on our manuscript.

Minor comments:

1) In Fig. 2c, I believe it would be useful to quantify the low-speed bundling assays rather than simply showing a single example gel. These tend to be fairly variable, and analyzing the percent F-actin pelleted across a few trials would provide confidence.

We have now added the requested quantification in Fig. 2D. The results show a clear difference in actin bundling activity between WT and optimized F-tractin

2) RE: Fig. 3: one of the important elements of a probe is its signal-to-noise ratio, which in this case would be the degree of apparent F-actin enrichment vs. cytosolic background. If this is similar between F-tractin_{opt} and WT F-tractin, it would be an important "selling point" for the optimized version. One way to measure this would be to quantify the images by making a mask from the phalloidin channel, then calculating the average F-tractin intensity within the mask vs. in the cytoplasm (excluding the nucleus). A scatterplot of these values for the different constructs, or comparing the average ratio of the cytoskeletal vs. cytoplasmic signal, would likely be informative. This analysis could likely be done with the authors' current data.

We thank the reviewer for this valuable suggestion. We have now quantified the SNR as suggested (Material and Methods, Fig. S3) and present the values in Fig. 3D. The results clearly show that WT and optimized versions of F-tractin behave very similarly (no significant difference) and provide comparable contrast to Lifeact. To further support the increased exchange rate of F-tractin_{opt} in FRAP experiments we have now also added the

quantification of nuclear enrichment for each probe (Fig. 3 E). This provides an indirect measure for actin affinity as only unbound probe molecules can enter the nucleus.

3) In Table S1, the authors list the helical rise as 28.9 Angstroms, but in the Methods the starting value is listed as 27.7 Angstroms (similar to what has been reported in many other F-actin structures). If the rise really jumped 1.2 Angstroms during IHRSR processing, this suggests there is likely a pixel size calibration issue.

We are very grateful to the reviewer for noticing this problem. Indeed, we found a pixel size calibration error. We therefore reprocessed the dataset and updated Fig. 1, Supplementary Table S1 and Supplementary Figure 1 accordingly. The corrected helical rise is now 27.4 Angstroms, which is similar to the previously reported values and to the starting value (27.7 Angstroms). The new reconstruction and the molecular model is available for review at https://drive.google.com/drive/folders/1Y_ahYC3eSHaQrBfaMBHleeJ6DNM0pLC2?usp=sharing

Suggestions / typos etc.:

4) I believe the potential impact of F-tractinopt is undersold. E.g. it is not really mentioned in the Abstract or Introduction: its potential significance as a useful tool could be emphasized a bit more.

We thank the reviewer for this suggestion. Indeed, the optimized F-tractin could be a useful tool in cell biology. We now mention the optimized F-tractin in the last paragraph of the introduction.

5) In the Methods, the authors first say they purified alpha-actin, but then in the co-sedimentation assays subsection they say they used beta-actin. If this is not a typo, the purification of beta-actin needs to be described.

We thank the reviewer for noticing this. All experiments were performed with alpha-actin. We corrected the typo.

6) Line 56: "setups" should probably be "systems"

Corrected.

7) Line 81-82: Could mention that the Lifeact binding site is similar here as well.

Changed as requested.

8) Fig. 1a: The contrast on the cryo-EM micrographs was very difficult to see when printed. I'd recommend adjusting the contrast, or low-pass filtering the images for display.

We adjusted the contrast of the cryo-EM micrographs on Fig. 1a as suggested by the referee.

9) Line 299: CTF "correction" should probably be "estimation"

Corrected.

Reviewer #2

In this study, Shatskiy and colleagues report a cryo-EM structure of the F-tractin:F-actin complex. The authors resolved 24 out of the 43 amino acids in the F-tractin peptide through their cryo-EM reconstructions. Their findings reveal that the N-terminal part of the F-tractin peptide, while not essential for F-actin binding, is crucial for its bundling effect. Additionally, they identified that the C-terminal helix exhibits interaction profiles similar to several actin-binding proteins. This study highlights that F-tractin and Lifeact comparably bind to the same F-actin sites. The structural insights provided here are instrumental in advancing the design of more effective F-actin probes.

This paper is well-written, conveying a clear and compelling message supported by their robust cryo-EM data. The methodology is appropriately presented. I believe this manuscript will engage the general readership of the JCB journal.

We thank the reviewer for the positive evaluation of our manuscript.

I have the following comments/suggestions:

1. To enhance the study, it would be beneficial if the authors included a discussion on the occupancy of F-tractin in cryo-EM reconstructions. Specifically, an explanation of the peptide's nature and how its solubility impacts its occupancy would provide valuable insights. Furthermore, clarifying how variations in occupancy align with natural binding behaviors could offer a deeper understanding of the interaction dynamics.

When processing the cryo-EM dataset, to improve the density corresponding to F-tractin, we performed alignment-free 3D classification with a focus mask to sort out actin subunits without F-tractin. Given that the classification retained ~50% of particles at this stage, we can estimate that F-tractin occupancy was ~50% (see Fig S1B). We now discuss this processing step in detail in the first paragraph of Results and discussion. However, we cannot directly relate this number to the nature or solubility of the peptide, nor can we discuss changes in occupancy due to interaction dynamics.

2. The authors also created an optimized F-tractin variant, which exhibits no bundling, which is a valuable contribution. Further, solving the complex structure of this optimized F-tractin in complex with F-actin could provide significant insights into bundling/binding. While this is not a hard request, such analysis would help identify essential residues within the C-terminus, distinguishing them from those that are dispensable, thereby enhancing our understanding of its structural and functional roles in binding vs bundling.

As requested by the reviewer, we determined the cryo-EM structure of the F-tractin_{opt} in complex with F-actin and present it in Fig. 2E and Figure S2. The new structure is almost identical to the resolved part of full-length F-tractin, except for the expected shorter C-terminus. This reinforces the notion that actin binding is not dependent on the N-terminal segment of F-tractin and that hydrophobic residues Val-22, Leu-25, and Phe-29 provide the main contacts of F-tractin to the actin filament.

3. The finding that F-tractin shares a common interaction interface with actin-binding proteins is intriguing. Strengthening the manuscript further, the authors could perform cellular imaging using the newly designed probes, particularly the optimized F-tractin, to demonstrate co-localization with the interaction partners (any) identified in Figure 4. Given that F-tractin is known to influence actin dynamics and may stabilize actin, it would be valuable to assess whether the novel probes developed in this study offer any functional advantages over native F-tractin.

This is a great suggestion. Unfortunately, colocalization experiments with cofilin in cells were not successful in our hands, as the Cofilin-GFP fusion protein remained completely cytosolic. In any case a simple competition as evaluated *in vitro* would be difficult to assess in cells, as most actin binding proteins act in concert with other factors and exhibit highly cooperative binding behaviors. This might ultimately be the reason for the low toxicity of actin binding peptides, which still allow essential activities of myosins and cofilin to actin filaments at moderate expression levels.

4. Authors should discuss a possible mechanism for F-tractin mediated bundling. Discuss how it contributes to bundling vs binding in the manuscript.

We thank the reviewer for this suggestion. We propose that the positively charged N-terminus of F-tractin non-specifically interacts with negatively charged residues on the surface of actin filaments, leading to actin bundling. We added a corresponding sentence in the Results and discussion section of our manuscript.

5. The authors have provided details on fixed cell preparation for imaging; however, it would strengthen the manuscript if they also included a description of the cell preparation protocols used for live cell imaging for conducting FRAP experiments.

Thanks for the suggestion. We have updated the details of cell preparation for the FRAP experiments in the Imaging section of Material and methods.

6. Expand the model building and refinement section in the methods part of the manuscript.

We expanded the model building and refinement section as requested by the reviewer.

Reviewer #3

The manuscript by Shatskiy et al entitled "Structure of the F-tractin-F-actin complex" describes the cryoEM structure solution of F-tractin peptide bound to filamentous actin. The authors have validated the structure using structure-function analysis and present a shorter F-tractin peptide called F-tractin-opt, which performs similar to the original peptide. The structure and biochemistry is of decent quality. However, it is hard to comprehend what new findings the authors are presenting, as the title aptly points this manuscript is a structure study and does not offer any insights to cell biologists to mitigate the issues with F-tractin as an F-actin probe or solutions. Therefore, I do not recommend publication in JCB, unless the authors add some new insights using their structure. Few major points are summarized below;

We thank the reviewer for the positive assessment of the quality of our study. We are sorry that the reviewer did not appreciate the utility of our work for the cell biology community. In fact, we believe that cell biologists will strongly benefit from our study for the following reasons:

1) Both Lifeact and F-tractin are popular actin-binding probes and researchers need to make a decision on which one to use. In fact, several groups have spent considerable time and resources (Flores Sci Rep 2019; Spracklen Dev Biol 2014; Belin Bioarchitecture 2014; Martins BioRxiv 2024) to identify differences between them. However, our high-resolution structural data, for the first time, shows that both probes exhibit very similar modes of interaction with F-actin.

2) F-tractin alters actin organization, disrupts cell migration and induces actin bundling (Belin Bioarchitecture 2014; Vosatka Plos One 2022). Based on cryo-EM and biochemical data, we propose mechanisms for these artifacts. This understanding will allow cell biologists to make informed decisions on the best actin marker for their experiments.

3) We have previously determined the cryo-EM structure of the Lifeact-F-actin complex (Belyy PLoS Biol 2020). We are happy to see that our structural data has inspired and motivated researchers to further modify the probe for various applications in cell biology, including optogenetics (Kroll et al., Nat Methods 2023), super-resolution imaging with the LIVE-PAINT approach (Bhaskar Protein Sci 2023), and proteomics-based analysis of actin-binding proteins (E. Emily Joo Meth in Mol Biol 2024). We expect that the new structure of F-tractin-F-actin will serve as a similar basis for the development of future actin probes.

1. Based on structure the authors show that N-terminus is dispensable for F-actin binding and reduces actin bundling. This region is then attributed to the artifacts seen in in vivo and cell studies that uses F-tractin. The authors should show these in vivo methods using F-tractin-opt, this will be a great application of their structure.

This is a great suggestion, but unfortunately difficult to address. In general, the reported cellular defects induced by weak-affinity actin side binding peptides like Lifeact, F-tractin and Utrophin are fairly mild and only occur at high expression levels of the respective markers. The observed effects were also very specific to the particular cell type used. Considering the low affinity for both the full length and optimized F-tractin variants we did not see or expect any serious alterations of cellular actin organization in our experiments. A more detailed and sensitive study would currently be beyond the scope of our study.

2. Similar to the above point, have the authors considered combining lifeact and F-tractin sequence information to produce an alternate, better probe.

We thank the reviewer for this suggestion. Indeed, our structural analysis suggests that combining the N-terminus of F-tractin and C-terminus of Lifeact might result in a probe with

a higher affinity to F-actin. Such chimeric constructs will certainly be an interesting direction to explore in the future.

3. Since F-tractin binds to similar region as lifeact and it has been shown that D-loop state affects lifeact binding it is not clear whether D-loop state affects the F-tractin binding. This should be discussed.

We thank the reviewer for this suggestion. We now added this information at the end of Results section.

4. The authors have used ADP-ribosylation assay as a method to show F-tractin interference; it is not clear what the authors trying to convey here. It should also be noted that, myosin/cofilin and other actin-binding proteins have a larger footprint and may have better binding properties than the TcART.

We chose TcART for the experiment presented on Fig. 4F precisely because it binds to F-actin with lower affinity than myosin or cofilin. This makes TcART a very sensitive probe to compare Lifeact and F-tractin interactions with F-actin. We now added this motivation to the text.

5. In general the discussion of the manuscript falls very short in many aspects, this needs a complete overhaul.

We modified the discussion part following suggestions of the reviewers.

6. The lifeact Kd indicated in line 207 refers to the author's earlier study. However, other studies have shown different Kd values. This needs to be corrected.

We thank the reviewer for raising this issue. We have now added the Kd value from Courtemanche Nat Cell Biol 2016. In two other sources (Riedl 2008 Nat Meth; Kumari EMBO 2020), the Kd of Lifeact was measured with pure Lifeact peptide rather than with the Lifeact fusion protein, making it not entirely comparable to our experimental setup.

7. Almost all the studies of F29A mutant has been done in the background of F-tractin-opt. The authors should show the mutant effect in original F-tractin peptide.

Indeed, our experiments with the F29A mutation were performed with an optimized version of F-tractin. However, our cryo-EM analysis performed with full-length F-tractin clearly showed that F29 is the major site of interaction between the peptide and F-actin. Therefore, it is very unlikely that the F29A mutation would yield any different results in the context of the full-length peptide.

January 8, 2025

RE: JCB Manuscript #202409192R

Alexander Belyy
University of Groningen

Dear Dr. Belyy:

Thank you for submitting your revised manuscript entitled "Structure of the F-tractin-F-actin complex". As you will see, reviewers are completely satisfied with this work and recommend publication. We would be happy to publish your paper in JCB pending final revisions necessary to meet our formatting guidelines (see details below).

A. MANUSCRIPT ORGANIZATION AND FORMATTING:

Full guidelines are available on our Instructions for Authors page, <http://jcb.rupress.org/submission-guidelines#revised>. Submission of a paper that does not conform to JCB guidelines will delay the acceptance of your manuscript.

1) Text limits: Character count for Articles is < 40,000, not including spaces. Count includes abstract, introduction, results, discussion, and acknowledgments. Count does not include title page, figure legends, materials and methods, references, tables, or supplemental legends.

2) Figures limits: Articles may have up to 10 main figures and 5 supplemental figures/tables.

3) Figure formatting: Scale bars must be present on all microscopy images, including inset magnifications. Molecular weight or nucleic acid size markers must be included on all gel electrophoresis. Please avoid pairing red and green for images and graphs to ensure legibility for color-blind readers. If red and green are paired for images, please ensure that the particular red and green hues used in micrographs are distinctive with any of the colorblind types. If not, please modify colors accordingly or provide separate images of the individual channels.

4) Statistical analysis: Error bars on graphic representations of numerical data must be clearly described in the figure legend. The number of independent data points (n) represented in a graph must be indicated in the legend. Statistical methods should be explained in full in the materials and methods. For figures presenting pooled data the statistical measure should be defined in the figure legends. Please also be sure to indicate the statistical tests used in each of your experiments (either in the figure legend itself or in a separate methods section) as well as the parameters of the test (for example, if you ran a t-test, please indicate if it was one- or two-sided, etc.). Also, if you used parametric tests, please indicate if the data distribution was tested for normality (and if so, how). If not, you must state something to the effect that "Data distribution was assumed to be normal but this was not formally tested."

** Please indicate whether t-tests were one- or two-sided in figure legends.

5) Abstract and title: The abstract should be no longer than 160 words and should communicate the significance of the paper for a general audience. The title should be less than 100 characters including spaces. Make the title concise but accessible to a general readership.

** In agreement with a previous request by Reviewer 1, we feel the optimized F-tractin is a central focus of this work and will draw interest from readers. Please include a sentence in the abstract describing this new probe.

6) Materials and methods: Should be comprehensive and not simply reference a previous publication for details on how an experiment was performed. Please provide full descriptions in the text for readers who may not have access to referenced manuscripts. We also provide a report from SciScore and an associate score, which we encourage you to use as a means of evaluating and improving the methods section.

7) Please be sure to provide the sequences for all of your primers/oligos, plasmids, and RNAi constructs in the materials and methods. You must also indicate in the methods the source, species, and catalog numbers (where appropriate) for all of your antibodies. Please also indicate the acquisition and quantification methods for immunoblotting/western blots.

8) Microscope image acquisition: The following information must be provided about the acquisition and processing of images:

- a. Make and model of microscope
- b. Type, magnification, and numerical aperture of the objective lenses
- c. Temperature
- d. Imaging medium

- e. Fluorochromes
- f. Camera make and model
- g. Acquisition software
- h. Any software used for image processing subsequent to data acquisition. Please include details and types of operations involved (e.g., type of deconvolution, 3D reconstitutions, surface or volume rendering, gamma adjustments, etc.).

10) Supplemental materials: There are strict limits on the allowable amount of supplemental data. Articles may have up to 5 supplemental figures. Please also note that tables, like figures, should be provided as individual, editable files. A summary of all supplemental material should appear at the end of the Materials and methods section.

13) ORCID IDs: ORCID IDs are unique identifiers allowing researchers to create a record of their various scholarly contributions in a single place. At resubmission of your final files, please provide an ORCID ID for all authors.

15) A data availability statement is required for all research article submissions. The statement should address all data underlying the research presented in the manuscript. Please visit the JCB instructions for authors for guidelines and examples of statements at (<https://rupress.org/jcb/pages/editorial-policies#data-availability-statement>).

Please note that JCB requires authors to submit Source Data used to generate figures containing gels and Western blots with all revised manuscripts. This Source Data consists of fully uncropped and unprocessed images for each gel/blot displayed in the main and supplemental figures. Since your paper includes cropped gel and/or blot images, please be sure to provide one Source Data file for each figure that contains gels and/or blots along with your revised manuscript files. File names for Source Data figures should be alphanumeric without any spaces or special characters (i.e., SourceDataF#, where F# refers to the associated main figure number or SourceDataFS# for those associated with Supplementary figures). The lanes of the gels/blots should be labeled as they are in the associated figure, the place where cropping was applied should be marked (with a box), and molecular weight/size standards should be labeled wherever possible. Source Data files will be directly linked to specific figures in the published article.

WHEN APPROPRIATE: The source code for all custom computational methods published in JCB must be made freely available as supplemental material hosted at www.jcb.org. Please contact the JCB Editorial Office to find out how to submit your custom macros, code for custom algorithms, etc. Generally, these are provided as raw code in a .txt file or as other file types in a .zip file. Please also include a one-sentence summary of each file in the Online Supplemental Material paragraph of your manuscript.

Journal of Cell Biology now requires a data availability statement for all research article submissions. These statements will be published in the article directly above the Acknowledgments. The statement should address all data underlying the research presented in the manuscript. Please visit the JCB instructions for authors for guidelines and examples of statements at (<https://rupress.org/jcb/pages/editorial-policies#data-availability-statement>).

B. FINAL FILES:

Thank you for your attention to these final processing requirements. Please revise and format the manuscript and upload materials within 7 days. If you need an extension for whatever reason, please let us know and we can work with you to determine a suitable revision period.

Thank you for this interesting contribution, we look forward to publishing your paper in Journal of Cell Biology.

Sincerely,

Bruce Goode
Monitoring Editor
Journal of Cell Biology

Tim Fessenden
Scientific Editor
Journal of Cell Biology

Reviewer #1 (Comments to the Authors (Required)):

The authors have very thoroughly responded to the reviewer comments, including my own. I think the paper is substantially strengthened, particularly by the additional quantification of the cellular imaging data. It should be accepted for publication in the JCB without further delay.

Reviewer #2 (Comments to the Authors (Required)):

The authors have addressed all my comments and suggestions.